# Development of an IoT-Based Construction Worker Physiological Data Monitoring Platform at High Temperatures

**DOI:** 10.3390/s20195682

**Published:** 2020-10-05

**Authors:** Jung Hoon Kim, Byung Wan Jo, Jun Ho Jo, Do Keun Kim

**Affiliations:** 1Department of Civil and Environmental Engineering, Hanyang University, Seoul 04763, Korea; kimj32@hanyang.ac.kr (J.H.K.); kpxjuno08@hanyang.ac.kr (J.H.J.); 2Research and Development Centre, Youngshine D&C, Gyeonggi-do 13487, Korea; kimdokeun@daum.net

**Keywords:** physiological data, monitoring, smart band, construction worker, fuzzy logic

## Abstract

This study presents an IoT-based construction worker physiological data monitoring platform using an off-the-shelf wearable smart band. The developed platform is designed for construction workers performing under high temperatures, and the platform is composed of two parts: an overall heat assessment (OHS) and a personal management system (PMS). OHS manages the breaktimes for groups of workers based using a thermal comfort index (TCI), as provided by the Korea Meteorological Administration (KMA), while PMS assesses the individual health risk level based on fuzzy theory using data acquired from a commercially available smart band. The device contains three sensors (PPG, Acc, and skin temperature), two modules (LoRa and GPS), and a power supply, which are embedded into a microcontroller (MCU). Thus, approved personnel can monitor the status as well as the current position of a construction worker via a PC or smartphone, and can make necessary decisions remotely. The platform was tested in both indoor and outdoor environment for reliability, achieved less than 1% of error, and received satisfactory feedback from on-site users.

## 1. Introduction

Globally, there has been increased concern regarding workers’ safety and health because of many cases of illness and accidents at work [1,2,3,4,5,6,7,8,9,10,11,12,13,14,15,16,17]. In the construction industry, which mainly takes place outside and requires high levels of physical activity, the number of casualties is the highest among all industries [18,19,20]. According to a 2018 annual report from the Korea Occupational Safety and Health Agency, both fatal and non-fatal accidents occurred the most in the construction industry (26.6% and 27.1%, respectively). These ranks have been maintained over the last 10 years [21]. Likewise, in the United States, the mortality rate in the construction industry showed an upward trend from 2011 to 2015; construction is considered to be one of the most hazardous industries [22].

In addition to accidents on the jobsite, Korean construction workers have high tendencies to suffer from occupational illnesses, such as heat-related illness due to their work environment. The Centers for Disease Control and Prevention (CDC) recognizes heat stress as an element of occupational illness. Certain industry sectors are at higher risk, including construction workers, who are often exposed to extreme heat or work in hot environments [23]. In fact, according to a heat-related illness surveillance in 2018 conducted by the Korea Centers for Disease Control and Prevention (KCDC), 4526 people suffered from heat-related illness (48 deaths), and 28.1% of these patients were engaged in outdoor work. Moreover, in Korea, an average of 503 reported cases of heat-related illness have occurred annually since 2011 [24]. Previous studies have found that construction workers in the United States have 13 times the risk of heat-related illness or death compared with other industries, and the increasing frequency of these illnesses presents growing concerns about the occupational safety of workers [8,25]. These statistics demonstrate the vulnerability to environmental hazards faced by construction workers and the need to provide effective solutions to reduce incidents.

Traditionally, construction workers’ health conditions have been measured by subjective methods, such as surveys. However, because of the nature of self-reported questionnaires, these measurements may be partially biased. Additionally, surveys typically take place during break times; therefore, they may not reflect a worker’s condition while they are active. Furthermore, stopping workers during their tasks in order to complete a questionnaire hinders their work and workers may respond carelessly [26]. Therefore, in order to improve the reliability of the test method, sensor-type continuous measurement is encouraged in the construction field, where physical demands vary depending on the time, task type, and work conditions [5]. Originally, wireless body area sensor networks (WBSNs) were developed for patients recovering from an incident and/or those with chronic diseases that required continuous health monitoring. The Occupational Safety and Health Administration (OSHA) defines heat strain as a body’s physiological response to heat stress; naturally, responses include an increased heart rate and sweating. When these responses function improperly, they lead to an elevated core temperature, which may cause heat-related illness or death [27]. Moreover, the International Standards Organization (ISO) has indicated that responses to heat strain include a loss of body mass through sweating, and an increase in body core temperature, skin temperature, and heart rate [28]. For reactions to physiological strain, the heart rate is mainly measured as it is the earliest response [29].

As sensor technologies have improved, attempts have been made to adopt WBSNs in a variety of other areas, including the construction industry. For example, WBSNs in construction sites can provide effective tools to locate a worker’s position in a large construction site and to track assets in real time. However, the transmitting coverage and accuracy, which can be affected by environmental objects, facilities, the human body, etc., can still be improved [12,14,30]. Although previous results have showed a high accuracy when identifying unsafe acts and conditions, some methods require multiple pieces of equipment. Additionally, in dynamic conditions, like at a construction site, technical obstacles are inevitable [31,32,33]. Recently, based on the fact that wristband-type wearable devices and smart watches can be used to locate one’s position or to check on their physiological status, multiple attempts have been made to replace traditional physiological test methods [1,2,4,5,34]. However, most studies related to the construction field have been conducted in lab conditions.

This platform collects a worker’s physiological data through a wearable armband that consists of three sensors. The three sensors of the armband are PPG, temperature, and accelerometer sensors. The PPG sensor measures the volume of blood flow by detecting changes in the intensity of the reflected light. The accelerometer sensor provides the current position of a worker, while the thermometer transmits the current skin temperature of a worker. The hardware consists of a microcontroller (MCU), GPS module, low-power wide-area network (LoRa) module, and power-supply, all of which are embedded into a single circuit board. Data from the MCU form a worker’s current physiological status, which is directly sent to the web and to an smartphone application via the LoRa network for visualization. We used a testbed (the Gyeongbu Expressway Straightening Project, Korea Expressway Corp and Han-ra Corp, Whasung City, Korea) for the performance evaluation.

## 2. Materials and Methods

### 2.1. Overview of the IoT-Based Physiological Data Monitoring Platform

The main purpose of the developed platform is to observe an individual construction worker’s physiological conditions under thermal environments. Mainly, it contains the following two parts: an overall heat assessment (OHS) and a personal management system (PMS). The OHS serves to protect field workers from potential heat-related illness by adopting a thermal comfort index (TCI) provided by the Korea Meteorological Administration (KMA). Rather than using a thermometer to determine the break time at a construction site, the TCI also considers the work environment. The TCI is evaluated based on the wet bulb globe temperature (WBGT), but it is modified to differentiate certain work environments and age groups. In the OHS system, the thermal environmental data are obtained from KMA via the internet and are evaluated based on the TCI. OHS is the basis for a manager’s judgment of whether to provide workers with rest or further actions.

PMS provides a method to monitor individual health status, for which local meteorological issues might not always be the main cause of the abnormality. PMS allows an off-the-shelf mobile armband to monitor individual worker’s conditions, and it transmits the results to a server through the LoRa network. The obtained data (i.e., GPS, skin temperature, and PPG sensor) from a worker, after the filtering process, become sources for determining the individual risk level based on fuzzy logic assessment. Based on the individual physiological data and the local meteorological data from the OHS, in real time, the developed platform warns if the TCI values indicate an issue or if a registered worker’s physiological data is considered abnormal. Finally, a construction manager is notified of these results via the web or their smartphone, allowing them to respond instantly so as to prevent a negative incident. Figure 1 shows the overall process of the platform.

### 2.2. Factors Used to Develop the Platform

#### 2.2.1. Overall Heat Assessment (OHS)

Even after 50 years of use, the wet bulb globe temperature (WBGT) is still widely used to manage occupational heat stress. The WBGT responds to crucial components of the climate, including the amount of sunlight, wind, air temperature, and humidity. However, the WBGT can only provide general guidelines for preventing the adverse effects of heat. Although many nations still accept this measurement because of its convenience, limitations and errors do exist [35]. The heat index used in our proposed platform, i.e., the thermal comfort index (TCI), is developed based on the WBGT of the Korea Meteorological Administration (KMA). This has been used domestically to manage occupational heat stress. TCI is an index used to prevent heat-related illness by developing an estimation model for various heat-vulnerable areas (e.g., farming areas and outdoor workshops), as well as for the general environment. TCI is achieved by calculating the correction value of each heat-vulnerable area based on the time difference between the values from an automated surface observing system (ASOS) and its WBGT values [36]. *TCI* is determined by Equation (1):(1)$$\mathrm{TCI}=\;-0.24418+0.553991T_{w}+0.455346T_{a}-0.00217\times \left(T_{w}{}^{2}\right)+0.002782T_{w}T_{a}$$
$$\begin{array}{ll} T_{w}=T_{a}\;\times & arctan(0.151977\times {\left(RH+8.313659\right)}^{\frac{1}{2}}+\arctan\left(T_{a}+RH\right)\\ & -\arctan\left(RH-1.676331\right)\\ & +0.00391838\times \left(RH^{\frac{3}{2}}\right)\times \arctan\left(0.023101\times RH\right)-4.686035 \end{array}$$
here, $T_{a}=drybulb\;temperature$, $T_{w}=wetbulb\;temperature$, and RH=relative humidity.

The *TCI* values are categorized into five stages along with a color on the main page of the monitoring web service, as shown in Table 1. As a safeguard for construction workers, OHS provides recommended strategies based on local construction sites. Color change according to the stage provides the field manager grounds for giving site workers rest. The flow of the OHS is shown in Figure 2.

Moreover, *TCI* weights certain values based on the target and working environment, depending on the time of day as Table 2 shows. In this study, weighting values for construction sites have been applied.

#### 2.2.2. Personal Management System (PMS)

Despite TCI providing useful measures to protect the working group from thermal hazardous, risk still exists as the physical status of workers is different. Here, in order to minimize the risk, rather than only applying OHS, a personal monitoring system was added. Figure 3 shows the overall flow of the system. The proposed system allows for a manager or a supervisor to receive an alert through the web or a mobile application if there is any indication of an abnormality among the registered workers. The server collects four types of data through the device (PWB-300, Partron, Hwaseong, South Korea): (a) the location of a worker, (b) a worker’s position within a three-axis area, (c) the skin temperature of a worker, and (d) the heartbeat rate of a worker. Physiological data along with one’s current location are monitored through the wearable device, which is located on the worker’s arm, and the server receives data via the LoRa network. The transmitted data become variables for determining the risk level after the filtering process. Among the four risk levels, PMS automatically sends an alert to a manager and HQ if it reaches “attention” level.

### 2.3. System Architecture

#### 2.3.1. Hardware

To collect workers’ physiological data while they are working, the authors selected an off-the-shelf wearable smart band. Figure 4 illustrates the components of the PWB-300 (Partron, Hwaseong, South Korea): the transmitter node, which consists of three sensors (PPG, Acc, and skin temperature); two modules (LoRa and GPS); and a power supply. These are embedded into an MCU. The receiver node includes the server, which processes data from the transmitter node and displays the results on a mobile phone application or in a web browser. Two nodes are connected via the LoRa network in order to provide a higher accuracy and larger coverage while also reducing costs. Previous studies have used GSM or Bluetooth as transmitting networks [6,10,35]. However, in construction sites, which typically include wide areas and environmental obstacles, the range of these technologies is limited to a few hundred meters. Alternatively, LoRa offers a range of a few to tens of kilometers [36]. The technical specification of the LoRa module is shown in Table 3.

STM32L476 (STMicroelectronics, N.V., Amsterdam, The Netherlands) was used for the MCU, and contains ARM 32-bit Cortex-M4 CPU (Arm Limited, San Jose, CA, USA) with an FPU with a 512 KB flash memory and 128 KB SRAM. A dynamic range of 100 dB with 10 samples per second (SPS) to 1000 SPS pulse frequency, AFE4404 (Texas Instruments, Dallas, TX, USA) was used for the PPG sensor, and LSM6DS3 (STMicroelectronics, N.V., Amsterdam, The Netherlands) measured the accelerometer. A skin temperature sensor (DTS201, Partron, Hwaseong, South Korea) with an accuracy of ±0.3 °C from 34.0 to 42.5 °C was applied, while a GPS module (TD1030, Taidou Microelectonics, Shenzhen, China) with a 3 m horizon and 4.5 m vertical accuracy was used. BQ24121 (Texas Instruments, Dallas, TX, USA) was chosen for the power supply, which uses a 3.7 V, 350 mAh, 502035 Li-ion polymer battery. It carried out 500 cycle battery life, which is higher than 80% of the initial capacities of the battery.

#### 2.3.2. Sensor Signal Filtering Process

As the device is located on the user’s arm, the movement of the user and their sweat can potentially cause an error in the measurements. Moreover, in a construction site, where a large number of noises and artifacts exists, artifact removal and error correction processes are necessary. For the PPG sensor, bandpass filtering and threshold filtering were applied. In order to capture useful data, a band filter within a range of 0.5–4 Hz was applied [37]. During the validation tests, the transmitted data, which indicate the position of a worker, were proven to be accurate to about 3 to 5 m, while the raw form of physiological data (skin temperature and PPG signal) provided little information about the workers’ physical status, and therefore a filtering or adjustment process was required (Figure 5.)

According to previous studies [38,39], heat stoke is characterized by a core temperature higher than 40 °C, and heat exhaustion often occurs when the core temperature falls between 38 °C and 40 °C. However, skin temperature varies in different body parts; therefore, the following core body estimation method [40] using skin temperature was required:(2)$$T_{core}=T_{skin}+\alpha \times \left(T_{skin}-T_{ambient}\right)$$

The authors used 0.7665 as the adjustment value in Equation (2), as shown in Table 4. Moreover, an outlier removal filter was applied to smooth the skin temperature signals. The results of the filter process are shown in Figure 6.

#### 2.3.3. Risk Level Evaluation Process

In order to build a platform that automatically determines the risk level of a worker, a risk level evaluation process was designed using fuzzy logic of MATLAB (MathWorks, Natick, MA, USA). Figure 7 shows the overall process of the fuzzy implementation process. Physiological information, including the heartbeat rate (H), core temperature (CT), work intensity (W), and time interval (T), were collected as the input variables, and risk level (RL) was used as the output, all of which is based on the PMS. The risk level is divided into four levels—safe, concern, attention, and danger.

As shown in Figure 8, in the input variables, the pi-type membership was chosen while the triangle membership function was used for the output. The heartbeat rate was divided into three fuzzy sets. The lower limit of the heartbeat rate was selected as 60, while the caution range was established based on the Karvonen method [41]. As measuring an individual’s maximum heart rate requires hours of testing, simple age-based formulas were established for the upper limit [42,43,44,45,46]. Among them, *HRmax* = 207 − (0.7 × *age*) was selected, because it reflects a wide range of ages (30–75 years) with high confidence interval bounds (±1 SD) [46]. The core temperature and time interval or duration were divided into four fuzzy sets based on previous works [47,48,49,50,51,52]. The work intensity was divided into three fuzzy sets based on the construction site occupational health management report from the Korea Occupational Safety and Health Agency [53].

As shown in Figure 9, the rule base was designed with “And”, “Or”, and “Not” logic, and was then processed to the Mamdani fuzzy method. Finally, by using the center of gravity method, the risk level output was obtained. Table 5 shows detailed values for the risk level evaluation process.

#### 2.3.4. Back-End Monitoring System

Each device contains its own MCU, which directly transmits data using the LoRa module in TCP/IP form; therefore, the IoT gateway is not required in this platform. The collected information is received by a back-end monitoring system where the workers’ health status is analyzed and displayed. The primary functions of the back-end monitoring system display the current stage of the overall risk level, recording the worker’s physiological information, GPS history, and environmental information. It also sends an alert to a manager and HQ if the individual’s risk level reaches “attention” level. Table 6 shows the specification for the platform server.

### 2.4. Platform Web Interfacing and Features

The system features and web monitoring page (Figure 10c), which feature simple tabs and visual data, was created to be accessible to field managers. The main page displays the current status of the OHS and PMS. Figure 10b shows the required fields for managers to input when they distribute the device to a worker. The web page shows a list of the workers, along with their picture, heartbeat rate, skin temperature, age, blood type, device number, and work type (Figure 10d). The server provides a graph for the current set of stored data for review. Furthermore, a user can access a worker’s data history, including their core temperature, GPS location, and heartbeat rate; users also have the option to change the graph to display information for a day, week, month, or year (Figure 10a). Physiological data transmitted by the smart band are transferred to either a smart phone or the web via LoRa. The web interface is able to provide access to both PCs and smart phones.

The main features are as follows:(1)Weather information: shows the current location’s weather data, including the thermal comfort index and the number of workers, sensors, incidents, and responses.(2)Workers’ status: shows the list of registered workers in the project and their locations. The physiological status of individual workers can also be observed.(3)Registration: a new worker can be registered. Personal information (including a worker’s task type and blood type) and the device registration can be made.(4)History: shows the history of the physiological data and the location of a worker.(5)Setting: allows a user to modify the interface of the page.(6)Help: shows a technical manual for managers.

## 3. Validation Tests and Results

To validate the developed platform, two types of validation tests were conducted—indoor and outdoor. The indoor test was conducted to compare the transmitted data from the device with the results from the graded exercise test (GXT), which is the most widely used method to study the relationship between exercise and physiological systems [54]. The measurements were recorded by increasing the slope by 2.5% every minute at a speed of 5.1 km/h, also known as the modified Balke protocol. Three subjects were tested with different devices in case of inferior products. The characteristics of the subjects are described in Table 7. The results shown in Table 8 are the average data of the three trials for each subject. The percentage of error between GXT and PWB-300 was found to be less than 1%.

The outdoor experiment was set up in a construction site. The performance evaluation was conducted on the Gyeongbu Expressway Straightening Project, Korea Expressway Corp and Han-ra Corp, Korea. The project was 4.70 km in length, including a 10-lane road in both directions, and was located in Wha-sung City, Korea (Figure 11). Physiological monitoring data were collected from 18 workers during the summer period of the project (June to August).

The goal of the study was to perform an initial implementation of the system so as to monitor the abnormality under high temperatures for workers in a construction site. Physiological monitoring data were collected for three months. During this period, 18 workers (ranging in age from their 30s to their 50s) were invited to participate. Prior to the test, the workers input their personal information (i.e., age, height, weight, blood type, and normal blood pressure) and the managers input the intensity of the work considering a worker’s daily tasks (Figure 10b). As shown in Figure 12, in the test bed, skin temperature measurements were taken using a Polygreen KI-8280 infrared thermometer (KI-8280, Seoul, Korea), and data from the PPG sensor were compared with measurements from a Rossmax automatic heartbeat monitor (BI701, Greencross Medical Science Corp., Yongin, Korea). The comparison results showed a 95.7% accuracy.

In some cases, the risk level in PMS indicated an “attention” level during the break time, while the TCI indicated the “concern” level. Onsite managers conducted interviews, and a lack of sleep and the previous night’s drinking were reported by those workers. The effects and/or correlations between a worker’s heartbeat rate and their sleep or alcohol consumption were not taken into account in this study. The preliminary results showed the reliability of the proposed platform in detecting abnormality based on physiological data via the LoRa network and by acquiring OHS in real time. Considering the nature of the system, it is important to conduct qualitative analyses based on user experiences; in this case, field managers were interviewed. The authors interviewed the field managers who operated the platform during the test period. Interviewees were satisfied with the features of the system, especially with regard to the ability to determine the individual risk level.

## 4. Conclusions

Construction workers are at high risk of exposure to high temperatures due to rising global warming and the demanding physical nature of their work. Therefore, in this study, a physiological data monitoring platform was developed to reduce heat-related risk in outdoor construction sites. By combining real-time environmental data with individual physiological data, the developed platform allows for monitoring a worker’s status and to manage it more effectively. The IoT-based physiological data monitoring platform is composed of two parts: overall heat assessment (OHS) and personal management system (PMS). The OHS separates users into five stages and suggests responsive actions based on the thermal comfort index (TCI), which represents the risk of heat-related illness; the TCI was developed by the Korea Meteorological Administration (KMA). TCI allows heat stress to be assessed in construction sites by weighting certain values based on the target and working environment, depending on the time of the day. PMS determines the individual physiological risk level based on fuzzy logic using acquired data from a commercially available wearable biosensor (PWB-300). Moreover, by applying the Karvonen method and maximum heartrate equation into the risk evaluation process, the platform efficiently manages individual physiological risk level. The device also transmits a worker’s data to the server via the LoRa network, which does not require smartphone connection and covers a range of a few to tens of kilometers. Transmitted physiological data can be accessed using a web browser or smart phone, allowing managers to detect and respond to abnormalities on a worksite. A performance assessment was conducted on both indoor and outdoor testbed. These results confirmed that this platform serves as an effective tool to monitor the physiological data of a worker so as to prevent incidents at a high temperature. The data used to support the results of this study are available from the corresponding author upon request.

## Figures and Tables

**Figure 1 sensors-20-05682-f001:**
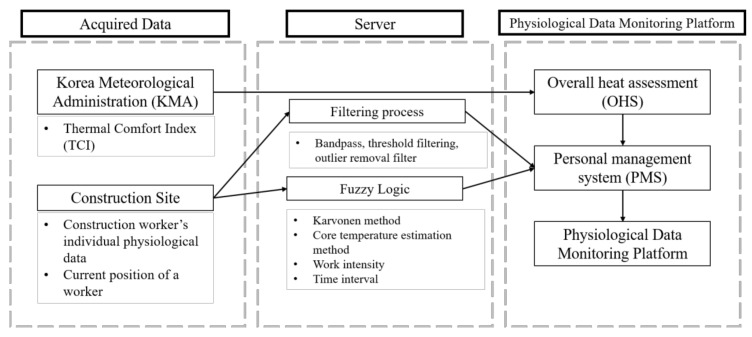
Overview of the physiological data monitoring platform.

**Figure 2 sensors-20-05682-f002:**
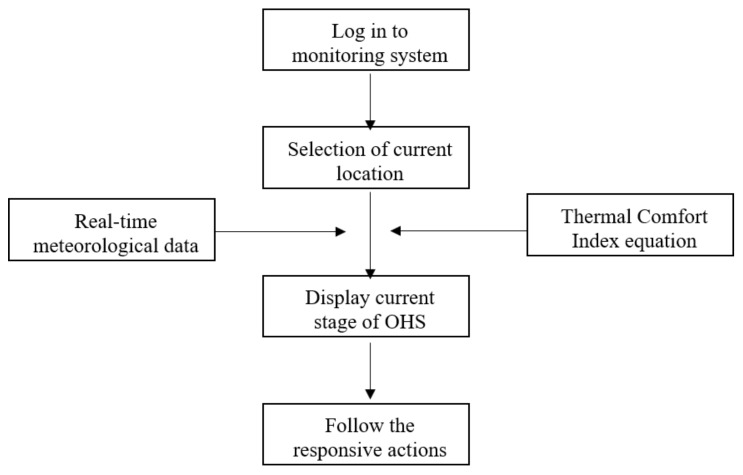
Flow of the overall heat assessment (OHS).

**Figure 3 sensors-20-05682-f003:**
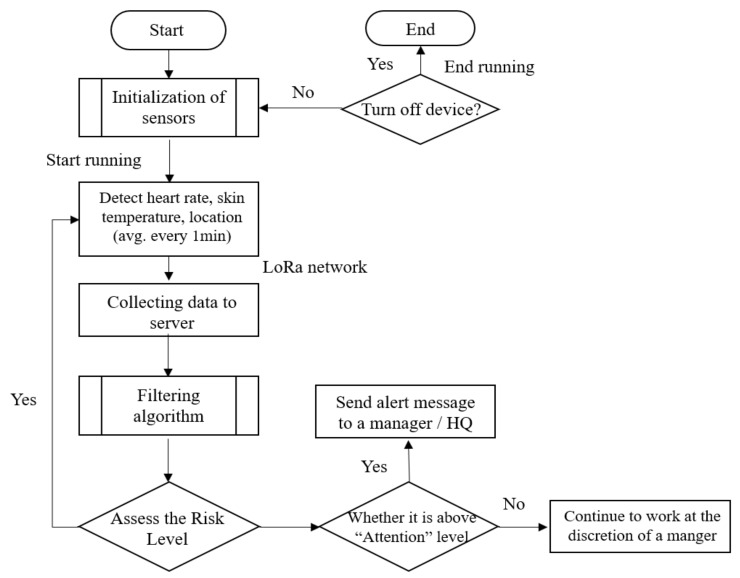
Flow of the personal management system (PMS).

**Figure 4 sensors-20-05682-f004:**
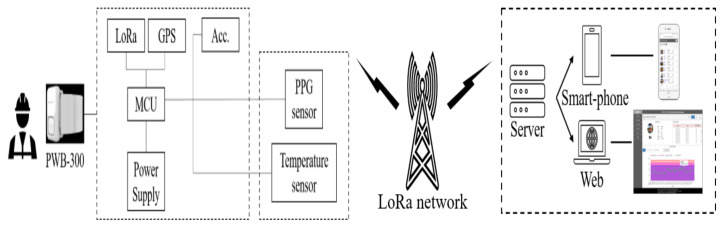
Hardware components of the platform.

**Figure 5 sensors-20-05682-f005:**
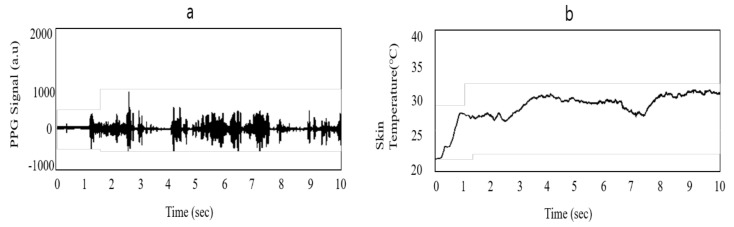
Raw form of signals from the PWB-300: (**a**) PPG signal and (**b**) skin temperature signal.

**Figure 6 sensors-20-05682-f006:**
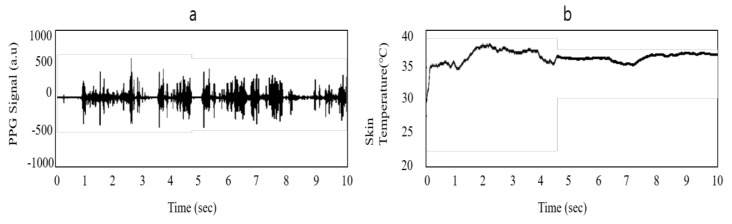
Filtered form of signals from PWB-300: (**a**) PPG signal and (**b**) skin temperature signal.

**Figure 7 sensors-20-05682-f007:**
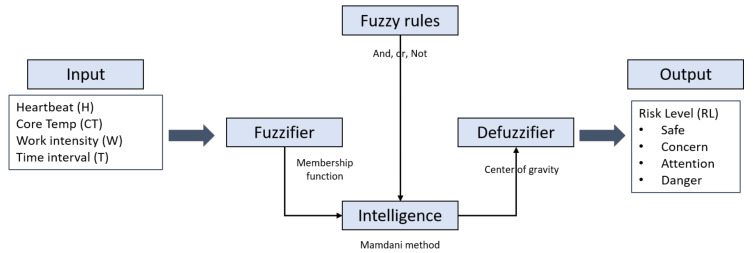
Fuzzy implementation process.

**Figure 8 sensors-20-05682-f008:**
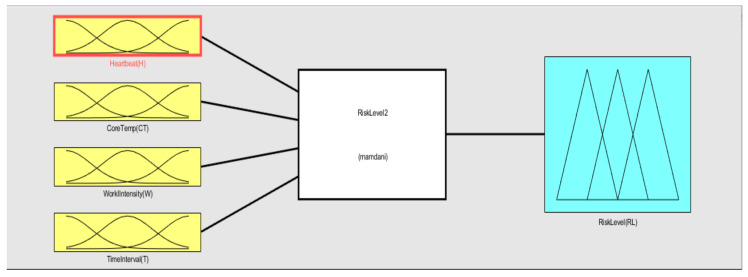
Membership function plot.

**Figure 9 sensors-20-05682-f009:**
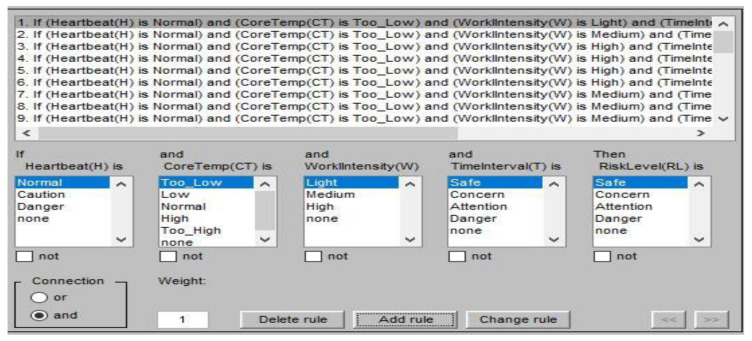
Fuzzy rule base.

**Figure 10 sensors-20-05682-f010:**
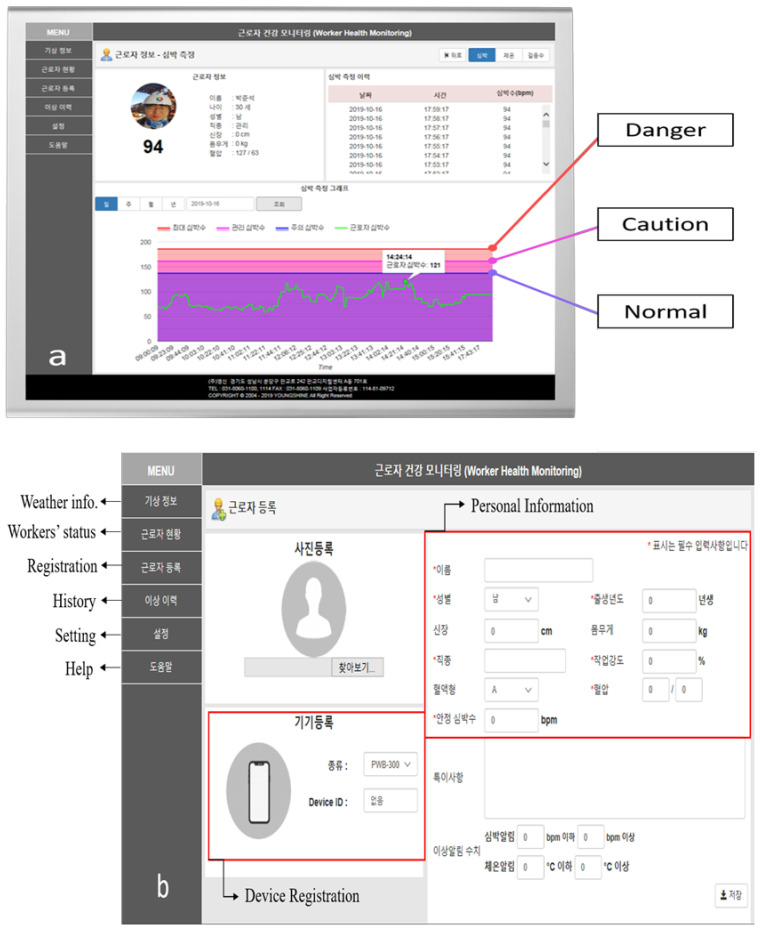
Web interface of the physiological monitoring platform: (**a**) Heartbeat rate monitoring page, (**b**) worker registration page along with the features of the monitoring platform, (**c**) the main page of the platform, and (**d**) physiological status of the registered workers in a site.

**Figure 11 sensors-20-05682-f011:**
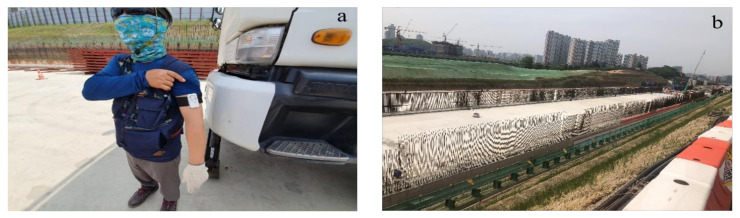
Performance test setup: (**a**) a worker with PWB-300 [55] and the (**b**) Gyeongbu Expressway Straightening Project, Korea.

**Figure 12 sensors-20-05682-f012:**
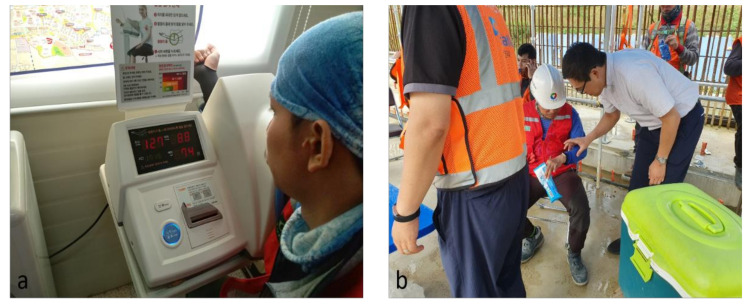
Outdoor validation test: (**a**) heartbeat comparison and (**b**) calibration of the device.

**Table 1 sensors-20-05682-t001:** Responsive actions and range values for each stage in an outdoor work environment, including road, construction site, and shipyard.

Stage	Range Value	Responsive Actions
Severe (Purple)	Above 30	All workers should stop working immediately and rest in a cool environment until further notice
Warning (Red)	30–28	All workers should preferably stop working as soon as it is possible
Caution (Orange)	28–25	Need to reduce work time or workload and take breaks frequently
Concern (Yellow)	25–21	Attention should be given to those who are vulnerable to heat dissipation
Attention (Grey)	Below 21	

**Table 2 sensors-20-05682-t002:** Thermal comfort index (*TCI*) weighting values relative to the time of the day (KMA).

Time of the Day (h)	Children	Heat Vulnerable Area	Farming Area	Greenhouse	Outdoor Workshops		
					Road	Construction Site	Shipyard
03	0	1.0	0	0	0.3	1.2	0
06	0	0	0	0	0.5	1.0	0
09	2.9	0.7	0.1	2.3	1.2	1.3	1.7
12	2.5	1.7	0.5	4.0	1.4	1.6	2.2
15	1.2	0.9	1.1	4.0	1.3	1.2	1.0
18	−0.9	0.3	0.6	2.1	0.9	0.7	0
21	0	5.0	0	1.0	0.5	1.4	0
24	0	2.5	0	0.5	0.3	1.2	0

**Table 3 sensors-20-05682-t003:** Technical specification of the low-power wide-area network (LoRa) module.

Specification	Value
Model name	SX1276 (SEMTECH, Camarillo, CA, USA)
Band	920–925 MHz
Data rate	0.18–37.5 kbps
Range	5 km (Urban), 15 km (Rural)
Band width	7.8–500 kHz
Sensitivity	−111 to −148 dBm

**Table 4 sensors-20-05682-t004:** Adjustment parameters for the core temperature in each body part [40].

Body Part	α
Rectal	0.0699
Head	0.3094
Torso	0.5067
Hand	0.7665
Foot	2.1807

**Table 5 sensors-20-05682-t005:** Risk level evaluation items and values.

Classification	Items		Value Range					
Input	Heartbeat (H)	Status	Normal		Caution		Danger	
		Range (bpm)	60 ≤ H ≤ 99		$\left(HRmax-Normal\;HB\right)\times \left(Work\;intensity\;\%\right)+\left(Normal\;HB\right)$		*Hrmax* ≤ H	
	Core Temperature (CT)	Status	Too low		Low	Normal	High	Too High
		Range (°C)	CT ≤ 32		33 ≤ CT ≤ 35	36 ≤ CT ≤ 38	38 ≤ CT ≤ 40	40 ≤ CT
	Work intensity (W)	Status	Light		Medium	High		
		Range	0.75		0.5	0.25		
	Time interval (T)	Status	Safe	Concern	Attention	Danger		
		Range (min)	T ≤ 5	5 ≤ T ≤ 14	15 ≤ T ≤ 29	30 ≤ T		
Output	Risk Level (RL)	Status		Safe	Concern	Attention	Danger	
		Range		0 ≤ RL ≤ 10	11 ≤ RL ≤ 20	21 ≤ RL ≤ 30	31 ≤ RL ≤ 40	

**Table 6 sensors-20-05682-t006:** Specifications of the platform server.

Items		Specification
Server hosting	Server	Lenovo SR530 (Lenovo, Beijing, China)
		CPU: Xeon Silver 4208 8core 85 W 2.1 GHz (Intel, Santa Clara, CA, USA)
		RAM: 16 GB (1 × 16 GB) Single Rank x4 DDR4-2933 (Hewlett Packard Enterprise, San Jose, CA, USA)
		HDD: 300 GB SAS 10 K 12 Gb * 2ea (Hewlett Packard Enterprise, San Jose, CA, USA)
	Performance (range)	10 Mbps (1 Gbps Uplink)
	Rack	1U (19” standard rack − H2100 × W600 × D1000)
	OS	Server, OS, DB
		Windows (IIS + MSSQL)

**Table 7 sensors-20-05682-t007:** The characteristics of the experiment subjects.

	Age (Years)	Height (cm)	Weight (kg)
Subject 1	32	172	77
Subject 2	41	168	73
Subject 3	50	176	76

**Table 8 sensors-20-05682-t008:** Indoor GXT results.

Subject 1	Rest	Stage 1	Stage 2	Stage 3	Stage 4	Stage 5	Stage 6	Stage 7	Stage 8
G.X.T.	85	109	113	119	126	140	153	165	176
PWB-300	85	108	114	119	125	140	152	165	176
**Subject 2**	**Rest**	**Stage 1**	**Stage 2**	**Stage 3**	**Stage 4**	**Stage 5**	**Stage 6**	**Stage 7**	**Stage 8**
G.X.T.	93	111	115	119	129	140	157	165	179
PWB-300	93	111	114	120	130	141	158	165	179
**Subject 3**	**Rest**	**Stage 1**	**Stage 2**	**Stage 3**	**Stage 4**	**Stage 5**	**Stage 6**	**Stage 7**	**Stage 8**
G.X.T.	83	112	116	122	131	143	156	168	180
PWB-300	83	113	116	122	132	143	156	168	180

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
