# Peer review of "Development of an IoT-Based Construction Worker Physiological Data Monitoring Platform at High Temperatures"

_sensors, 2020, doi:10.3390/s20195682_

Round 1
Reviewer 1 Report
The paper describes an IoT-based data monitoring platform specifically designed and implemented with the purpose to monitor the performance of construction workers under high temperature. The main contribution of the authors is the design and implementation of a single circuit board able to elaborate data provided by a commercial device in order to evaluate the physiological status of workers.
The topic addressed in the paper is certainly interesting but I find few scientific links with the journal selected for publication. As can be seen from the content, the physiological data are acquired through a commercial device, processed using known filtering techniques and transmitted to the decision-making platform.
The way in which parameter anomalies are reported is also very simplistic (for example in the classification of categories of workers based on heartbeat).
Moreover, it would be advisable to test on a greater number of subjects and in different environmental conditions given the characteristics of the platform
For all these reasons I recommend reviewing the entire content of the paper, describing the sensor part in more detail, highlighting the advantages and disadvantages of the proposed solution. Furthermore, it would be appropriate to propose an intelligent algorithm for monitoring the health status of workers in the analyzed context (for example using data fusion techniques or fuzzy logic)
Some minor issues:
- Avg heart rate every 1 minute is sufficient for detect anomaly?
-
Add a reference for PWB-300 device
-
Improve the quality of Figure 7 (consider to add a separated figure for each considered interface)
Reviewer 2 Report
The paper is more-less of engineering type. From the point of view of signal processing, it very weakly describes the detection of heart rate and related signal filtering.
Validation also should be extended because indoor test was evaluated by statistical tests and outdoor test lacks an analytical side.
Further recommendations and notes:
Line 45: May be missign word "were": "patients were engaged",
Line 77: Period preceding lowercase letter,
Lines 101-102: Unclear meaning of the sentence,
Line 104: Hearbeat data mentioned. However only the PPG sensor was mentioned above
Table 1: The lines in last column are not visually separated. Right bottom cell overflows into row below.
Figure 3: Text overflows from entities. The "Follow the instruction" box is not described.
Figure 7: The screenshots have low image quality.
Table 8: The table can be replaced by a shorter verbal description.
Round 2
Reviewer 1 Report
I carefully analyzed the revisions made to the original paper, I believethat all the suggestions given were used to obtain an article with
greater scientific value. In particular, I notice a significant improvement in the methodology
used to determine anomalies based on the fuzzy logic process
(with the addition of paragraph 2.3.3). In conclusion, I believe that the paper, in its current version,
can be published with minor revision.
Reviewer 2 Report
All my minor requests for revision were well implemented.
Althought in my opinion the offered device and method have not been sufficiently verified, the article can be accepted as engineering work.